# Learning algorithms allow for improved reliability and accuracy of global mean surface temperature projections

Ehud Strobach [1,2]* & Golan Bel [3,4]*

Climate predictions are only meaningful if the associated uncertainty is reliably estimated. A standard practice is to use an ensemble of climate model projections. The main drawbacks of this approach are the fact that there is no guarantee that the ensemble projections adequately sample the possible future climate conditions. Here, we suggest using simulations and measurements of past conditions in order to study both the performance of the ensemble members and the relation between the ensemble spread and the uncertainties associated with their predictions. Using an ensemble of CMIP5 long-term climate projections that was weighted according to a sequential learning algorithm and whose spread was linked to the range of past measurements, we find considerably reduced uncertainty ranges for the projected global mean surface temperature. The results suggest that by employing advanced ensemble methods and using past information, it is possible to provide more reliable and accurate climate projections.

[1] Earth System Science Interdisciplinary Center, College of Computer, Mathematical, and Natural Sciences, University of Maryland, College Park, MD 20740, USA. [2] Global Modeling and Assimilation Office, NASA Goddard Space Flight Center, Greenbelt, MD 20771, USA. [3] Department of Solar Energy and Environmental Physics, Blaustein Institutes for Desert Research, Ben-Gurion University of the Negev, Sede Boqer Campus, 84990 Negev, Israel. [4] Center for Nonlinear Studies (CNLS), Theoretical Division, Los Alamos National Laboratory, Los Alamos, NM 87545, USA. *email: strobach@umd.edu; bel@bgu.ac.il

Climate predictions and projections are typically based on global circulation models, which simulate the multi-scale processes that form the climate system[1]. The different parameterizations of unresolved processes[2] and the lack of precise initial conditions[3] result in uncertain projections. Therefore, the quantification of the associated uncertainties is important[1,4]. One of the simplest methods, which is also commonly used in climate research, is to establish an ensemble of simulations by varying some of the uncertain factors and/or model characteristics (initial condition, parameterization, numerical schemes, grid resolution, model parameters, etc.)[5–9].

Ensembles can be used to generate probabilistic forecasts, in which the full probability distributions of the variables of interest are estimated, rather than focusing on the mean and/or the variance of the variables. Probabilistic forecasting was done for different climate variables and using different weighting methods for the ensemble members[4,10]. Ensembles representing the uncertainties in model parameters[10,11], the uncertainties due to differences between models[4,12–14], and the uncertainties of empirical statistical models[12,15] were used to generate probabilistic forecasts. Many of these studies and others[16] also considered models in which different initial conditions of the models were included in the ensembles, and methods were suggested to account for the similarity between different realizations of the same model or different generations of the same model[17–19].

The underlying assumption of conventional probabilistic forecasts is that the ensemble spread represents (or at least provides a meaningful sampling of) the actual uncertainty regarding the climate dynamics[20]. However, structural uncertainties, due to different representations of physical processes (or the inadequate or missing representation of processes in some or all the models), affect the ensemble spread[20,21] and the relation between the ensemble spread and the uncertainty associated with its forecast[5,22]. Nevertheless, various methods relying on the ensemble spread were used to assess the uncertainty[1,23–26]. The true uncertainty (which includes both the ontic and epistemic uncertainties), or the relation between the ensemble spread and the real climate system uncertainties, can only be derived from past observations[27].

The quality of an ensemble forecast should be measured by two characteristics: the obvious one is its accuracy (often quantified by the magnitude of the errors), and the second one, which is often overlooked, is its reliability. The reliability is the correct quantification of the probability of the occurrence of different ranges of conditions[28–30]. Specifically, we refer to reliability as the accurate quantification of the fraction of observations within different confidence level intervals (which is equivalent to comparing the cumulative probability distribution). In order to evaluate probabilistic predictions, measures accounting for the reliability were developed and tested[31–34].

It was argued and shown that weighting climate models according to their ability to correctly simulate current and past conditions can reduce the structural uncertainties[1,4,35–38]. Evaluating climate projections by their past performances has been questioned, because the projections simulate future conditions that may be significantly different from past and current conditions, and certain limitations have been pointed out[1,20,39–42]. However, numerous studies have shown that this approach is legitimate and indeed improves the quality of climate predictions and projections[1,17–19,43–48].

Recently, a new method for the quantification of the uncertainties associated with ensemble predictions was suggested[49]. The method is based on studying the relation between the spread of the ensemble member predictions (quantified by the ensemble standard deviation (STD)) and the ensemble root mean squared error (RMSE). Obviously, this approach requires simulations of past conditions, which allow the calculation of the RMSE. The most general method of those suggested[49] is the asymmetric range (AR) method, which relies only on the assumption that the relation between the ensemble spread and the error does not change significantly with time (i.e., the relation found during the learning period remains the same during the prediction/projection period). The prediction of an ensemble is the distribution of the climate variables or their anomalies. The AR method has the advantage of estimating independently the range of likely conditions above the mean and the range of conditions below the mean (in this sense, it is asymmetric). The AR method was shown to improve the reliability of surface temperature and surface zonal wind prediction by an ensemble of CMIP5 (coupled model intercomparison project, phase 5) decadal simulations[49]. The improvement of the reliability demonstrated the validity of the assumption that the relation between the ensemble spread and its error does not vary considerably for decadal predictions.

Climate projections are not expected to be synchronized with the natural variability of the climate system (e.g., we do not expect the correct timing of future El Niño events). However, they are expected to be synchronized with the climate system responses to changes in its atmospheric composition[1,50]. Some synchronization between the simulations and observations can be found in the historical part of the CMIP5 projections (Supplementary Fig. 1). This synchronization can be attributed to the forcing by the observed atmospheric composition (which mostly varies by the greenhouse gas emissions and large volcanic eruptions; e.g., 1992–1993 cooling related to the Mount Pinatubo eruption[51,52], the effects of the Agung eruption in 1963, and the El Chichón eruption in 1982). The climate system responded to the volcanic eruptions within several years. These relatively short response times, which are reflected by the changes in the global mean surface temperature (GMST) within the period used for the evaluation of the model performances, suggest that comparing the model projections with observations may be used to assess the model performances in simulating the climate system responses to external forcing.

In this study, we use a sequential learning algorithm for weighting the members of an ensemble of CMIP5 climate projections and the AR method in order to study the relation between the spread of the weighted ensemble and the errors of its projections. Combining the two techniques we show that the uncertainties of future climate projections are significantly reduced relative to those specified in the last IPCC report[1]. Various tests that include different validation periods and the application of the methods to specific model projections, rather than the reanalysis data, demonstrate the robustness of the combined methods.

## Results

**Study design and performance tests**. We use an ensemble of CMIP5 projections[50]. The simulated GMST 20-year running averages for the period of 1967–2100 are considered (the annual average GMST for the period 1948–2100 is used to construct the 20-year running average time series). The value assigned for each year represents the average GMST of the 20-year period ending on that year. Running averages could be used in this study because our methodology does not assume that sequential values of the variable are independent. The first stage in the analysis involves weighting the ensemble members according to their past performances (during the learning period of 1967–2016; total of 50 simulated and observed values of the GMST 20-year average; note that the first independent point is the value assigned to the year 2036, which does not include any year that was used during the learning process), using the exponentiated gradient average

(EGA) sequential learning algorithm[45,46]. The weighting of the models is done during the learning period in a sequential manner. At each time step (i.e., for each observation revealed), the weights are updated in order to bring the weighted mean of the ensemble closer to the observations. Moreover, in order to ensure stability of the weights, the maximal allowed change for the weights is limited. The weighting method is expected to improve the weighted ensemble average forecast[45,46] by finding the optimal combination of weights such that the weighted ensemble mean is as close as possible to the observed value during a learning period. It is important to note that the EGA is designed to track the observations and not the best model. This in turn enables its outperforming the best model in the ensemble[45,46]. In previous works, we tested several other learning algorithms. We found that all the sequential learning algorithms that we tested performed better than the simple average (equally weighted ensemble) and better than the linear regression. Among the sequential learning algorithms, the EGA, which attempts to maximize the mutual information between the forecast and the observations[53], performed better (smaller RMSE and better estimated spread). The weighting affects not only the ensemble mean (the projection of the EGA forecaster) but also the ensemble's STD, which is often used to quantify its spread and the uncertainties associated with the projection. In the second stage of the analysis, the relation between the weighted STD and the projection errors is established using the AR method[49] in order to estimate the range of likely GMST values at different periods. The AR method calculates a pair of time-independent correction factors for each desired confidence level (confidence level $c$ implies that the extreme higher and lower tails of the probability distribution, each with a probability $(1−c)/2$, are excluded). The two correction factors ($\gamma_u(c)$ and $\gamma_d(c)$), multiplied by the time-dependent ensemble STD ($\sigma_t$), are then used to determine the uncertainty range for each confidence level: one for the range above the ensemble mean and one for the range below the ensemble mean (see the Methods section). By doing this, the AR method can generate an asymmetric interval for selected confidence levels.

The performance of our methodology was tested by splitting the learning period into learning and validation periods. A 35-year learning period (15-year validation period) was found to be long enough to reduce the projection error by 64%, to decrease the 0.9 confidence level uncertainty range by 65%, and to be more reliable (see also the Methods section and Supplementary Fig. 1). A reversed experiment, in which the last part of the period for which there is reanalysis data was used for learning and the first period was used for validation, also demonstrated improvement relative to the simple average. In addition, out-of-sample tests were performed in which one model was removed from the ensemble and used as the projected variable. In these tests, the EGA+AR also showed improvement relative to the simple average (see full details in the Methods section and Supplementary Fig. 2).

**Global mean surface temperature uncertainties**. Figure 1 shows the GMST uncertainties for the different representative concentration pathway (RCP) scenarios. Compared to the simple averages (Supplementary Fig. 3), the uncertainty ranges in this figure are considerably reduced by using the learning process (learning the relation between the ensemble spread and its error). The uncertainty range of the 0.9 confidence level is found to be 68–78% smaller than the range calculated using an equally weighted ensemble and assuming a Gaussian distribution of the ensemble projections. Similar ratios, between the uncertainty ranges estimated using the EGA and AR methods and those estimated using the equally weighted ensemble and the Gaussian

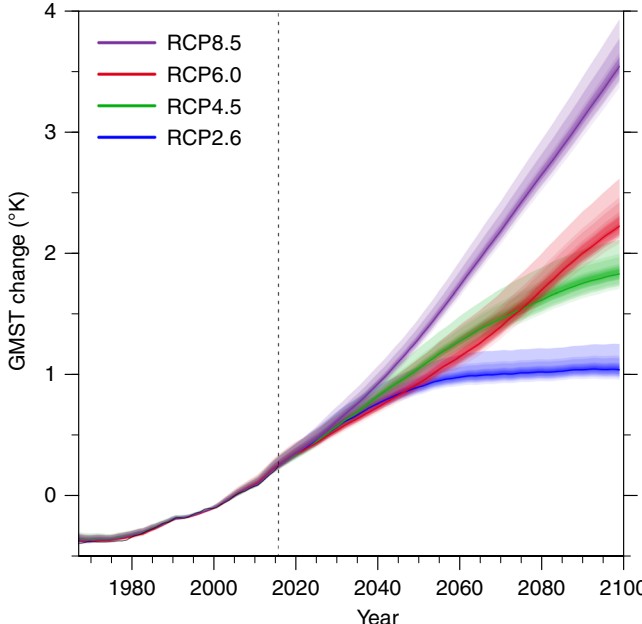

**Fig. 1 Projections of global mean surface temperature.** The 20-year running average of the global mean surface temperature (GMST) change relative to the 1986–2005 average for the representative concentration pathways included in the coupled model intercomparison project, phase 5. The thick lines represent the weighted ensemble mean for the 20-year running average GMST projections, and the shadings represent different significance levels (from 0.1 to 0.9) of the associated uncertainty (based on the exponentiated gradient average weighted ensemble and the asymmetric range estimation of the uncertainty ranges). Black lines represent the national centers for environmental prediction (NCEP) reanalysis. The left part of each panel (to the left of the dashed vertical line) represents the learning period, and the right part of each panel represents the validation period.

assumption, are also found for other confidence levels (see Supplementary Table 1; numerical values of the ranges estimated using the two methods can be found in Supplementary Tables 2 and 3).

We also find that the distribution of the GMST 20-year average is highly asymmetric. For example, the higher significance levels are skewed toward higher than the mean values, and for the 0.9 confidence level, the $\gamma_u$ values are even more than twice as large as the $\gamma_d$ values (namely, the range above the ensemble mean including 45% of the probability is more than twice as large as the range below the mean, which includes the same probability). The values of $\gamma_u$ and $\gamma_d$ for different significance levels and RCPs are given in Supplementary Table 4. In Supplementary Table 5, we provide the skewness and the excess kurtosis of the distribution of the 20-year average GMST. As can be seen, for all RCPs, the skewness does not vanish and is positive (implying that the distribution is right-tailed), and the excess kurtosis is positive, implying a distribution for which rare events are more likely than in a Gaussian distribution. These results demonstrate that the AR method is not only capable of reducing the uncertainty ranges but is also capable of extracting the deviations from a Gaussian distribution and, therefore, provides more accurate and reliable estimates of the GMST probability distribution.

**The role of EGA and AR in reducing the uncertainties.** There are two elements that may influence the estimated uncertainty ranges using the EGA and AR methods: the EGA weights, which affect the ensemble STD (weighted STD vs. simple STD), and the

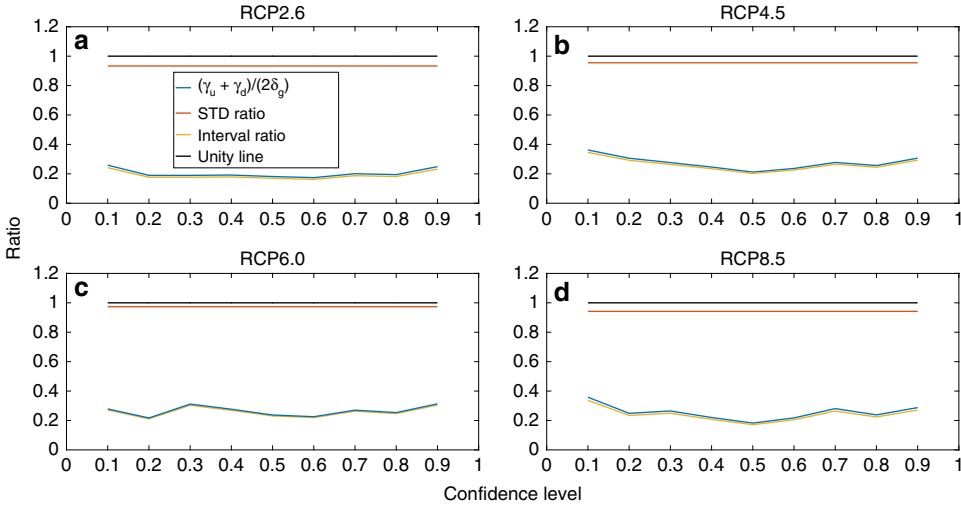

**Fig. 2 Reduction of uncertainties using the asymmetric range method.** The ratio between the confidence intervals of the asymmetric range (AR) and Gaussian methods for different confidence levels. The difference in the intervals stems from the different standard deviations (STDs) of the exponentiated gradient average (EGA) weighted and equally weighted ensembles and also from the correction factors due to the non-Gaussian distribution of the ensemble projections. The orange line presents the ratio between the EGA and equally weighted ensemble STDs (note that it differs between the representative concentration pathways (RCPs) due to the different ensembles and weights); the yellow curve represents the ratio between the sum of the AR correction factors ($\gamma_u(c) + \gamma_d(c)$) and the expected sum of the Gaussian distribution correction factors for a given confidence level ($2 \cdot \delta_G(c)$); and the blue curve represents the total uncertainty reduction, namely, the ratio between the uncertainty ranges estimated using the EGA and AR methods and those estimated using the equally weighted ensemble and the assumption of a Gaussian distribution of the ensemble projections, for different confidence levels. The black curve represents the unity line and is displayed for reference. **a–d** correspond to the denoted RCPs.

AR method correction factors, which multiply the STD and provide the relation between the STD and the uncertainty range for different confidence levels. The first element is time-dependent, and the second is constant during the projection period (see the Methods section for a more detailed discussion). We find that the average (2020–2099) EGA weighted STD is similar to the average STD of the equally weighted ensemble (the orange lines in Fig. 2). This suggests that the EGA learning does not converge to one specific model but rather spreads the weights among different models. Specifically, we find that only the MRI-CGCM3 has a considerably larger weight than the others, but most of the models were assigned a non-negligible weight (see Supplementary Tables 6 and 7). The main uncertainty reduction is due to the AR method ($\frac{\gamma_u(c) + \gamma_d(c)}{2 \cdot \delta_G(c)} < 1$; where, $2 \cdot \delta_G(c)$ is the correction factor corresponding to a Gaussian distribution). We also find that this reduction is true not only for the temporal average (2020–2099) but also for the entire time series of the projections (see Supplementary Fig. 4).

**Comparison of estimated probability distribution.** The PDs (probability distributions) of the GMST in different years differ due to the temporally varying STD, $\sigma_t$, and mean, $p_t$, of the ensemble of GMST projections. Note that the mean serves as the prediction of the ensemble, and both the mean and the STD are based on the weights assigned to the ensemble members. In Fig. 3, we present the probability distribution of the change in the 20-year average GMST, relative to the NCEP (national centers for environmental prediction) reanalysis 1986–2005 average, for two different periods and for the four RCPs included in the CMIP5. All RCPs predict significantly warmer GMST in the future, which is indicated by the separation of the 1986–2005 average GMST probability distribution from the probability distributions of the 2046–2065 and 2080–2099 average GMSTs. As expected, the uncertainties also increase with the increased lead time of the projections (indicated by the broadening of the distributions, see also Supplementary Tables 8 and 9). We also note that the

larger the assigned change in the greenhouse gas concentration, the larger the growth of the uncertainty. Relative to the ranges provided in the last IPCC report[1], our estimates of the uncertainty ranges are considerably reduced for all the scenarios (Supplementary Tables 8 and 9).

**Discussion.** Uncertainties in climate projections are of great importance for policy makers and practical applications including the development of adaptation and mitigation plans. The adequate quantification of the uncertainties and their reduction, where possible, are also at the core of climate dynamics research. The fundamental assumption underlying our methods is that it is legitimate to use our knowledge regarding past conditions and the simulations of these conditions in order to learn the relation between them.

Dividing the historical period into learning validation periods revealed that for the conditions in the last century, the assumption is valid. The combination of the EGA learning algorithm (to weight the models) and the AR method (to extract the relation between the ensemble spread and the actual uncertainties associated with the weighted ensemble projection) resulted in a considerable reduction of the uncertainties for all the RCPs and for the entire period of the projection. The reduction for the 20-year averages reached over 80%. Moreover, the entire probability distribution was derived by considering different confidence levels. A comparison of the CMIP5 ensemble spread with past observations clearly reveals an underconfident ensemble projection, suggesting that the actual uncertainty associated with these projections may be even smaller than estimated here. The method suggested and applied here was shown to be suitable for ensembles whose spread is dominated by model variability (rather than internal variability), which was found to be the case for multi-model ensembles of climate predictions and projections[9,21,54,55]. For ensembles of weather predictions where the internal variability is dominant, it was shown that there is a weak relation between the ensemble spread and its error[56].

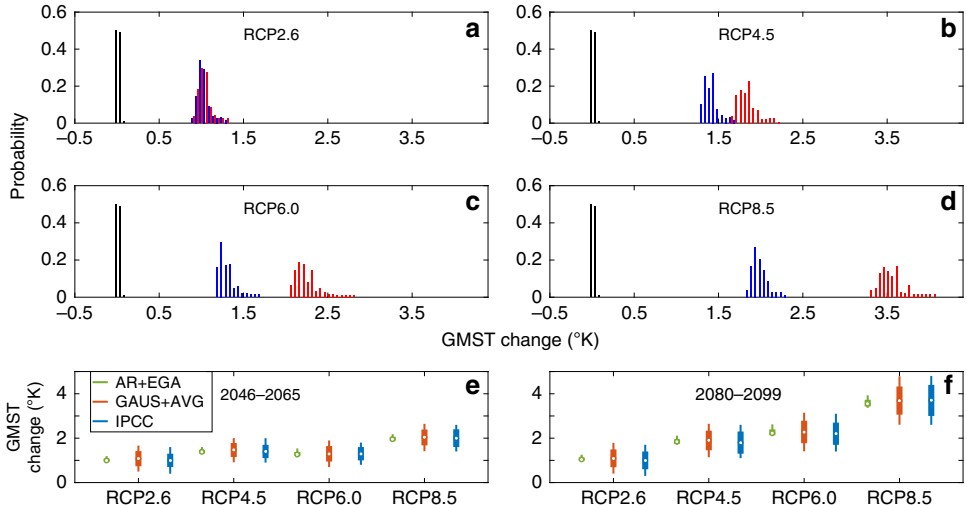

**Fig. 3 Probability distributions of the surface temperature change. a–d** The global mean surface temperature (GMST) change probability distributions of the 2046–2065 average (blue) and the 2080–2099 average (red) for the four representative concentration pathways (RCPs) included in the coupled model intercomparison project phase 5, relative to the 1986–2005 national centers for environmental prediction (NCEP) reanalysis average GMST (black). **e, f** Box plots of the GMST change relative to the 1986–2005 NCEP reanalysis average. The circles, boxes, and error bars represent the ensemble mean, the 16.7–83.3% uncertainty range, and the 5–95% uncertainty range, respectively. The blue, red, and green colors correspond to the international panel on climate change (IPCC) estimate, the estimation based on the equally weighted ensemble and the Gaussian assumption (GAUS+AVG), and the estimation based on the exponentiated gradient average and asymmetric range methods (AR+EGA), respectively.

We propose that the methods used here should find wider application in the analysis of ensembles of climate predictions.

## Methods

**Models and data.** We use all available surface temperature projections from the CMIP5 data portal. Projections of some of the models that were included in the last IPCC assessment report were not available when this study was initiated. Several other models were excluded from this study either because of incomplete data for the simulated period spanned in this study or because the cell area information was missing (for the calculation of area-weighted global means). Supplementary Table 6 lists the models that were included in the ensembles considered in this study and their scenario availability (different ensembles were constructed for each RCP based on the data availability). The internal variability of climate projections was found to be small compared with the model variability[54], and therefore, we decided to use one realization (initial condition) per model (the first listed in the CMIP5 data portal). The performance of each model was assessed by comparing the simulated GMST with the NCEP reanalysis data[57], which was considered here as the true value. The NCEP reanalysis was chosen here to represent the true value because it is a widely accepted reanalysis product and because of its relatively long record starting from 1948; the JRA55 reanalysis[58] was also tested but the fact that this reanalysis is shorter, thereby only allowing a shorter learning period, resulted in considerably larger uncertainties. We also tested observation-based products such as the HadCRUT4[59] and GISTEMP v3[60] global temperature datasets. These products resulted in similar uncertainty ranges as the NCEP, and therefore, they are not shown here. The simulations used in this study spanned the period between 1948, the first year for which NCEP/NCAR reanalysis data are available, and 2100. Both simulated results and the NCEP/NCAR reanalysis were time averaged to 20-year running averages (the value for each year in the time series corresponds to the average of the year and the 19 preceding years' GMST, thereby providing a time series from 1967 to 2100).

**The weighting of the ensemble models.** To weight the climate models, we use the exponentiated gradient average (EGA) algorithm[45,46,53]. The inputs to the EGA algorithm are time series of simulated GMST from an ensemble of forecasting experts (the climate models) and a time series of the projected variable (observations; in this study, we use the NCEP/NCAR reanalysis data as the observations). The EGA algorithm compares the simulated results from each of the ensemble members with past observations (using a squared error metric in our study) and weights the models based on their past performances. As a preliminary step, we bias-corrected the CMIP5 model outputs by subtracting from each model its temporal average during the learning period (1967–2017) and adding to it the NCEP reanalysis temporal average for the same period. The input to the EGA algorithm is, therefore, bias-corrected CMIP5 projections and NCEP reanalysis data. The output of the EGA at the end of the learning period is a weight for each model in the ensemble (Supplementary Table 7 shows the resulting weights for each model and RCP scenario). The original method was modified to ensure that

there are no large fluctuations in the weights during the learning period and that the learning rate is optimal[45,46]. The weighting procedure allowed us to derive the weighted ensemble STD, which in turn was used to derive the relation between the ensemble spread and the actual uncertainty range based on the asymmetric range (AR) method[49]. The comparison of simulated results with an observation-based product in this study does not suggest that we are trying to predict the future natural variability of the climate system (such as El Niño events); rather, it suggests that we weight the models based on their ability to correctly simulate the response of the climate system to observed forcing fluctuations over longer time scales. It is worth noting that repeating the same analysis using the annual averages rather than the 20-year averages also resulted in reduced uncertainties. However, in order to avoid criticism of the use of annual averages that are not expected to be synchronized with the simulated dynamics, we focus here on the 20-year averages.

**Constructing the PD of the GMST.** The AR method uses the past errors (relative to the NCEP/NCAR reanalysis) and ensemble STD to construct the PD of the GMST by multiplying the time-dependent (EGA or equally weighted) ensemble standard deviation (STD) with two optimized, significance-level-dependent correction factors that are time-independent. There are two correction factors for each significance level, one for the upper (above the mean) side of the PD ($\gamma_u(c)$) and one for the lower side (below the mean) of the PD ($\gamma_d(c)$), to allow for non-symmetric PDs to be captured. The upper and lower correction coefficients, $\gamma_{u,d}(c)$, are calculated after learning the fraction of the number of observations within a specific range $(p_t - \gamma_d(c) \cdot \sigma_t) - (p_t + \gamma_u(c) \cdot \sigma_t)$ during the learning period. The values of $\gamma_{u,d}(c)$ are chosen to be the smallest values that satisfy the conditions that at least a fraction $c/2$ of the observations are inside the area spanned by the two time series $[p_t, \gamma_u(c) \cdot \sigma(t)]$ during the learning period and similarly the fraction of observations within the area between the two time series $\gamma_d(c) \cdot \sigma(t)$, $p_t$ is $c/2$. Mathematically, these conditions are described by the following equations:

$$\gamma_u(c) = \inf\left\{\gamma_u \in \Re_{>0} : \frac{1}{n}\sum_{t=1}^{n}(\Theta[(p_t + \gamma_u \cdot \sigma_t) - y_t]) \geq \left(\frac{1+c}{2}\right)\right\} \quad (1)$$

$$\gamma_d(c) = \inf\left\{\gamma_d \in \Re_{>0} : \frac{1}{n}\sum_{t=1}^{n}(\Theta[y_t - (p_t - \gamma_d \cdot \sigma_t)]) \geq \left(\frac{1+c}{2}\right)\right\} \quad (2)$$

In the above equations, $\Theta(x)$ is the Heaviside step function ($\Theta(x) = 1$ for $x > 0$, $\Theta(x) = 0$ for $x < 0$, and $\Theta(0) = 1/2$), $p_t$ is the weighted ensemble average (forecasts for time $t$), $\sigma_t$ is the weighted ensemble STD at time $t$, $y_t$ is the observed (true) value at time $t$, and $n$ is the number of time points (length of the time series used) in the learning period. For example, $\gamma_{u,d}(c)$ should both be equal to the *probit* function ($\delta_G = \sqrt{2}erf^{-1}(1-c)$), if the observed distribution of the error is unbiased, and Gaussian. We repeated this process for multiple significance levels between 0.1 and 0.9. The resolution of the derived PD depends on the number of observations with values higher and lower than the ensemble projection during the learning period. The optimization of the correction coefficients $\gamma_{u,d}(c)$ can be done by including or excluding at least one observation, and this in turn limits the PD resolution to

$1/N_{u,d}$ (where $N_{u,d}$ are the numbers of observations larger and smaller than the projection, respectively; in this study, it was $N_u = 24$ and $N_d = 26$) in the above and below mean sides, respectively. Note that according to eqs. 1 and 2, the actual confidence level might be higher than the desired one due to the finite resolution. The AR method was developed initially for decadal climate predictions in which it was assumed (and verified) that the uncertainty correction coefficients ($\gamma_{u,d}(c)$) show only small fluctuations during the learning and the validation periods.

**20-year average probability distributions**. The above analysis results in the time series of the mean and the STD (based on the EGA weighted or equally weighted ensembles) and the values of $\gamma_{u,d}(c)$ for a range of desired confidence levels. As outlined in eqs. 1 and 2, for each confidence level, the excluded tails on both sides of the mean are equal (the integral over each of the tails is $(1−c)/2$). Considering the quantities above allows one to construct the full probability distribution. The AR algorithm outputs are ranges as a function of probabilities. To convert to probabilities as a function of ranges, we drew $10^7$ values from the derived distribution of the 20-year average GMST of 2065 and 2099. The depicted PDs of the 20-year average GMST are the histograms of the $10^7$ values. We verified that the probability distribution converges for this number of realizations (the differences were below any statistical significance).

**Testing the EGA and AR performance**. We test the performance of our methodology by dividing the 50 year period of 1967–2016 into two periods: a learning period and a validation (prediction) period. We tested different combinations of learning and validation periods, and we found that in order to improve the forecast (in terms of accuracy and reliability), we need >30 years of learning. In Supplementary Fig. 1, we show the forecast from 35 years of learning (1967–2001) and 15 years of a validation experiment (in all CMIP5 projections, the assigned atmospheric composition for the historic part until 2005 is based on observation, and later on, it depends on the RCP. We used the RCP 4.5 ensemble. The validation period of 2002–2016 includes years with RCP- rather than measurement-based atmospheric compositions in the last 12 years (2006–2017); it is worth mentioning that the variability between the different scenarios was found to be small compared with the model and internal variabilities during the first years of the projections[54]). We find that the RMSE of the equally weighted ensemble is larger than that of the EGA weighted ensemble (0.094 °C for the simple average compared to 0.034 °C for the EGA weighted average); in addition, using the AR method for estimating the uncertainty ranges resulted in smaller future uncertainty ranges for the EGA weighted ensemble and more reliable predictions (Supplementary Fig. 1). We also performed a reversed experiment, in which the first period was used for validation and the second period was used for learning. This test enables the use of periods with different trends in the learning and validation periods. We find that in this case, 40 years of learning (1977–2016) were needed to improve the accuracy and the reliability of the EGA+AR forecast during the validation period, 1967–1976 (relative to the equally weighted ensemble with the Gaussian assumption forecast).

For the equally weighted ensemble and the assumption of a Gaussian distribution of the ensemble projections, we found that the uncertainty range was much larger than the range expected from a reliable ensemble (the projections using these methods represented an underconfident forecast). A forecast based on the EGA weighted ensemble combined with the AR method was found to be close to reliable (see also Supplementary Fig. 1). Due to the better performance of the EGA weighted ensemble combined with the AR method, we focus on the results of this methodology.

In order to test the performance of the suggested method for projections, out-of-sample tests were performed, in which one model was removed from the ensemble and used as the projected variable during the 1967–2016 learning period. This type of test allows an additional test of the suggested method over a longer period. The weakness of the test is the fact that it uses a time series which is different (also statistically) from the one that the models aim to project. Some of the models in the CMIP5 ensemble share major components with other models and, as a result, have similar projections[47,61,62]. To perform a fair test, we removed similar models from the ensemble that was used in each of the tests (as performed in ref. [47]; see also Supplementary Fig. 5). In addition, by definition, the weighted ensemble (including an equally weighted ensemble) cannot project values outside the range spanned by the model projections. Therefore, we excluded from the out-of-sample test the two models with the warmest projections and the two models with the coldest projections (i.e., the extreme models were not used as projected variables). We find that the learning process (EGA+AR) reduces the RMSE and the uncertainty range relative to the simple average and the Gaussian assumption (the results are summarized in Supplementary Fig. 2). The EGA+AR projection is also significantly more reliable for the RCP8.5 and not significantly more reliable for the other three scenarios; here, we use the mean squared error of the deviation of the reliability curve from the line of identity as a reliability score. It is important to note that even for the scenarios in which the reliability is not significantly improved, the uncertainties are significantly smaller while the reliability does not worsen. The Error-Spread proper score[33] was also calculated, and according to this score, the EGA+AR method projections were significantly more reliable for the RCP2.6, RCP4.5, and RCP8.5 (results not shown). The full projection results of these tests with the CCSM4 as the projected variable are presented in Supplementary Fig. 6 and with the CSIRO-Mk3-6-0 as the projected variable in Supplementary Fig. 7. All

scores were calculated for the projection period, 2040–2099 (avoiding the first 20 years, which are dependent on the training data).

## Data availability
The CMIP5 model data that support our findings can be downloaded from The Program for Climate Model Diagnosis and Intercomparison (PCMDI) archive at https://esgf-node.llnl.gov/projects/cmip5/. The NECP reanalysis data are available from the National Oceanic and Atmospheric Administration (NOAA) Earth System Research Laboratory (ESRL) Physical Sciences Division (PSD) website at https://www.esrl.noaa.gov/psd/data/gridded/data.ncep.reanalysis.html.

## Code availability
Code generating figures and processed data will be available upon request.

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

## Acknowledgements
We are grateful to Samara T. Bel for proofreading the manuscript. We acknowledge the World Climate Research Programme's Working Group on Coupled Modelling, which is responsible for the CMIP, and we thank the climate modeling groups (listed in Supplementary Table 6 of this paper) for producing and making available their model output. For the CMIP, the U.S. Department of Energy's Program for Climate Model Diagnosis and Intercomparison provides coordinating support and leads the development of software infrastructure in partnership with the Global Organization for Earth System Science Portals.

## Author contributions
E.S. and G.B. contributed to the design of the research and the writing of the manuscript. E.S. performed the analysis and generated the figures.

## Competing interests
The authors declare no competing interests.
