## [Peer Review File · Nature Communications]

Reviewers' comments:

Reviewer #1 (Remarks to the Author):

This well-written, clear paper applies an interesting new methodology to the task of estimating uncertainty in climate projections. The new methodology, in effect a sophisticated ensemble calibration technique, may well be a significant improvement in the context of decadal climate prediction (where the authors have previously applied it) and perhaps also in shorter-term prediction contexts (e.g. weather forecasting). There, its value for calibration could be empirically tested out-of-sample.

What is much less clear to me, however, is that it makes sense to apply the methodology in the context of longer-term climate projection. As the authors note: "The fundamental assumption underlying our methods is that it is legitimate to use our knowledge regarding past conditions and the simulations of these conditions in order to learn the relation between them." They seem to mean: we assume that there exists, and we can identify, an intrinsic relationship between simulations from particular models and the observations or, more narrowly, a relationship that will continue to hold between them over the next century or so. This key assumption is not given sufficient justification. The authors note that: "Dividing the historical period into learning validation periods revealed that for the conditions in the last century, the assumption is valid." But this does not provide very strong evidence for the assumption, since clearly part of the challenge with quantifying uncertainty in longer term climate projections is that we may have reasonable doubts that the relationship between simulations and observations in the past century *is* going to be highly similar to the relationship in the future. This could be because of model tuning to past data, knowledge that some processes and feedbacks that may amplify in future are absent from models or only poorly represented, etc.

Such factors lead the IPCC WG1 to *augment* ensemble results with expert judgment when reaching conclusions about the uncertainty associated with projections of future climate change; they inflate the uncertainty estimate generated by the sort of methodology that this paper aims to improve upon (e.g. ensemble average plus Gaussian fit). This paper, by contrast, implicitly assumes that this sort of augmentation/inflation is not necessary. Some stronger argument needs to be given for this, beyond the performance on past data. While reducing uncertainty is a laudable goal, we want to have good reason to believe we're not reducing it artificially – given, as the authors note, the important decisions that may be influenced.

Reviewer #2 (Remarks to the Author):

The paper presents a study of a methodology to analyse ensemble projections with global climate models. It includes a method to generate a weighted mean of the ensembles and a method to link the ensemble spread to the uncertainties of the model mean. The weights are determined by using a sequential learning algorithm which minimizes the difference between the observations and the model mean. The relation between ensemble spread and the uncertainties is determined by a set of correction factors that are multiplied on the ensemble spread.

The methodology has been used before by the same authors in studies of decadal predictions. Here it is shown to produce reduced uncertainty estimates for the future climate projections.

The topic is important and the paper includes some interesting results.

However, I have some questions about the methodology that the authors should consider before I can recommend that the paper is accepted.

Major comments:

1) Validation: The validation period consists of 15 years following the training period. In this validation period the models have trends not too different from their trends in the training period.

This choice might give too optimistic results in the validation period. The authors should try to train the model on the period 1983-2017 and validate it on 1967-1982. Trends are different in these two periods and the results of the validation might change. Actually I don't understand why only data from 1967 are used. Why not use data from 1948 which would allow for longer training and validation periods?

Perhaps some kind of cross-validation -- using one model as the truth -- could be used for validation.

The sequential learning algorithm seems quite complex and it is not obvious what the algorithm actually learned. For example, does the last part of the training period influence the weights more than the first?

It would also be interesting to see if the weights reflect how well the individual models reproduce the observed trend in the training period. The authors should also compare the results to a simpler method where the weights are determined by simply minimizing the mean square error over the training period.

2) I am somewhat confused about the influence of the "bias correction". It seems that observed and modelled temperatures are all relative to their 1986-2005 averages. This means that the ensemble spread is low in this period. How would the results change if this interval was chosen differently? The CMIP5 historical experiments include all of the 20'th century so there are many choices available.

3) The error on the model mean consists of both a bias and a variance. It is not obvious to me if the Asymmetric Range method includes the bias.

More generally, the relation between ensemble spread and the ensemble mean error has been studied in details in weather forecasts (see, e.g., the discussion in Christiansen 2019, doi: 10.1175/MWR-D-18-0211.1). There it seems that the relation is in general weak. Why should we expect it to be different for climate projections?

Minor comments:

page 3: Reliability could be better described.

page 4: "These relatively short .. ". Why is it important here that the response times are short?

It would be nice if observations were also shown in Fig. 1 and Extended Data Figs. 1 and 2.

Page 14: p_t should be defined where it is first used.

Reviewer #3 (Remarks to the Author):

The submitted paper details a weighting methodology, presented here for time-series data, with

the goal of producing calibrated forecasts from a multi-member ensemble of physical models. The method is applied to climate model data from the CMIP archive, producing a calibrated forecast for future scenarios with a significantly smaller uncertainty range than would be obtained by considering ensemble mean and standard deviation.

Unfortunately, I am not confident in the calibrated projections - for a number of reasons I detail below. The approach - which may have value in short-timescale initialized forecast problems does not address the key uncertainties present in longer timescale projections from a structurally diverse ensemble of models. The literature review also doesn't reflect the large amount of literature discussion on the topic of CMIP model weighting (and its limitations) over recent years. As a result, I cannot recommend this paper for publication in Nature Communications.

The primary issue is related to the key assumptions - that there exists a time-independent multiplicative correction factor for the ensemble mean which applies in long term projections and that a model with smaller errors in global mean forced temperature response during the late 20th century is more reliable for century-scale projections than another model with larger errors. These statements are not self-evident, and have not been sufficiently demonstrated in an out-of-sample sense in this paper.

Firstly, the real world is a single realization of internal variability (as the authors acknowledge), but the mitigating approach used (moving average 20 year means) is not sufficient to eliminate this problem - some modes of variability will project onto the 20 year moving average and cannot be cleanly separated from the forced response. Thus, an assumption that a single realization is the forced response will lead to bias in the weighted mean projection.

Secondly, even if the forced response was known - low errors in the historical record do not imply low errors in the future. The historical response is a combination of responses to changing greenhouse gas and other anthropogenic and natural forcings. A model might get the right answer for the wrong reasons if its greenhouse gas and aerosol sensitivities are both too large - such a model would have a low historical error but would be unreliable in the future (e.g. Kiehl 2007). The ratio of greenhouse gas and aerosol forcing in 2100 under RCP8.5, for example is vastly different to the present day.

Similarly, recent warming is a function of both radiative feedback response to forcing and ocean heat uptake - so errors in these terms can also compensate and lead to the observed historical response in a model with compensating biases in climate sensitivity and ocean heat uptake (e.g. Mauritzen 2017).

These types of uncertainties (unlike for decadal predictions) are the first order uncertainty in future projections.

As such, representing the error around the mean as a time-invariant term is a strong assumption which has not been validated in the paper - and leads to significant overconfidence in the projected distributions. Rather, we would definitely expect the error term conditioned on global mean temperature alone to be time invariant, and to grow rapidly in the future - if we have conditioned the models based on recent climate, we have less reasons to trust our conditioning as the mix of forcings and rate of forcing change diverge from current values.

Splitting the time-series into recent history, and pre-1995 periods is insufficient. The regime in terms of forcing mix, and rate of change of forcing is too similar during the training and validation periods and so the test is not out of sample.

The key tests which the authors would need to perform for an approach such as this (or any weighting strategy) would be an out of sample validation using 2100 temperatures from CMIP models as truth and predicting their future evolution using the remaining ensemble. This would allow an semi-objective test of confidence in an out of sample test (with the caveat that models

with large similarities need to be removed from the comparison to make the test fair e.g. Sanderson 2017).

Kiehl, Jeffrey T. "Twentieth century climate model response and climate sensitivity." *Geophysical Research Letters* 34.22 (2007).

Mauritsen, Thorsten, and Robert Pincus. "Committed warming inferred from observations." *Nature Climate Change* 7.9 (2017): 652.

Sanderson, Benjamin M., Michael Wehner, and Reto Knutti. "Skill and independence weighting for multi-model assessments." (2017): 2379-2395.

APA

Reviewer #4 (Remarks to the Author):

I strongly support the eventual publication of this paper; it makes several points that are important and underappreciated both in the climate community and, just as importantly, in the community of numerate practitioners (say, economists) who wrongly assume just how much the properties of the model ensemble mean and ensemble standard deviation tell them. The paper also has some significant ambiguities or errors, I believe the paper should be appear after this a mended. I would strongly advise the editor NOT to reject this paper even give a highly negative review by another reviewer, please allow at least one revision before making a decision.

Wider Comments

The paper focuses on RMS and SD of GMT; it is now common to attempt fully probabilistic forecasts of annual GMT (Smith, L., Suckling, E., Thompson, E., Maynard, T. and Du, H. (2015) 'Towards improving the framework for probabilistic forecast evaluation', *Climatic Change*. I suggest that the authors note the probabilistic evaluation literature, and say (a few times) that exactly what they are doing (in scope and in evaluation).

Perhaps the major items missing in the paper is the role of structure model error (see Smith, L.A. (2002) 'What might we learn from climate forecasts?', *Proc. National Acad. Sci. USA*, 4 (99): 2487-2492. and recent citations thereof) This paper is cited by the AR4 (pg 797) which suggests the climate distributions from ensembles are too narrow, as structural model error has not been properly taken into account. What assumptions to the authors make to bypass Smith (2002)'s claim?

The two paragraphs above do not request more calculations, rather they require clarifying the assumptions required to apply the author's methods. This clarification is one the author strive for in their introduction and conclusion.

Defining the target as the 20 year average restricts out of sample forecasting in this century to one point in the near term (2020) with the next independent point at 2040.

I would accept the paper after these minor corrections. I would ask to see and harsh negative reviews of this paper in any event.

Small detail:

pg 2 projection and prediction are NOT defined by their lead time but by what they are conditioned on, they are both Predictions (see IPCC glossary)

"The prediction of an ensemble is its weighted average ..." pg 3. in general this is simply false. Ensembles from their beginning (Lorenz and Tennekes) aimed to quantify variability as distributions, ideally probability forecasts; it is known that ensemble spread is a function of the initial model state. In climate models the mean state itself varies hugely within each CMIP (hence the taking of anomalies).

pg 3 "much more reliable" please specify exactly what "more reliable" means (there are technical definitions for probability forecasts which are NOT what I think the authors mean) and which variables you claim it for.

pg 4 Please clarify what is meant by "synchronized".

pg 4 Please clarify that by running averages the average is defined purely into the past.

pg 10 first full paragraph starting "Uncertainties in ..." this is an important paragraph.

Reviewer #1:

This well-written, clear paper applies an interesting new methodology to the task of estimating uncertainty in climate projections. The new methodology, in effect a sophisticated ensemble calibration technique, may well be a significant improvement in the context of decadal climate prediction (where the authors have previously applied it) and perhaps also in shorter-term prediction contexts (e.g. weather forecasting). There, its value for calibration could be empirically tested out-of-sample.

*What is much less clear to me, however, is that it makes sense to apply the methodology in the context of longer-term climate projection. As the authors note: The fundamental assumption underlying our methods is that it is legitimate to use our knowledge regarding past conditions and the simulations of these conditions in order to learn the relation between them. They seem to mean: we assume that there exists, and we can identify, an intrinsic relationship between simulations from particular models and the observations or, more narrowly, a relationship that will continue to hold between them over the next century or so. This key assumption is not given sufficient justification. The authors note that: Dividing the historical period into learning validation periods revealed that for the conditions in the last century, the assumption is valid. But this does not provide very strong evidence for the assumption, since clearly part of the challenge with quantifying uncertainty in longer term climate projections is that we may have reasonable doubts that the relationship between simulations and observations in the past century **is** going to be highly similar to the relationship in the future. This could be because of model tuning to past data, knowledge that some processes and feedbacks that may amplify in future are absent from models or only poorly represented, etc.*

*Such factors lead the IPCC WG1 to **augment** ensemble results with expert judgment when reaching conclusions about the uncertainty associated with projections of future climate change; they inflate the uncertainty estimate generated by the sort of methodology that this paper aims to improve upon (e.g. ensemble average plus Gaussian fit). This paper, by contrast, implicitly assumes that this sort of augmentation/inflation is not necessary. Some stronger argument needs to be given for this, beyond the performance on past data. While reducing uncertainty is a laudable goal, we want to have good reason to believe were not reducing it artificially given, as the authors note, the important decisions that may be influenced.*

Authors' Response: We thank the reviewer for the thoughtful comments.

We are concerned that our statement, “The fundamental assumption underlying our methods is that it is legitimate to use our knowledge regarding past conditions and the simulations of these conditions in order to learn the relation between them,” was misunderstood. In this statement, our intention was to clarify that the only assumption we made is that knowledge of past observations and model simulations for the same past period can be used in order to learn the relations between the models and actual conditions. This fundamental assumption is the basis for any scientific work. The reviewer claims that the IPCC augments its uncertainty estimates by inputs other than the ensemble spread, and that the reason for the large uncertainties is the inflation of the ensemble spread due to these inputs. We wish to clarify that what we cite as the IPCC estimate for the uncertainties are the values derived from the ensemble that we considered, with the Gaussian assumption, and that these values are identical (up to any meaningful precision) to those reported by the IPCC. Therefore, this aspect of the reviewer’s comments is unclear to us. Moreover, any expert input has to be based on observations and/or experimental evidence. We used all the observations that are considered reliable.

We also wish to cite Smith [2002a], who shows that the ability to reproduce past observations is a necessary condition for any model. In order to have confidence about the ability of the models to produce accurate estimates of future conditions, additional knowledge is needed regarding the structural uncertainties of the models. It was also argued that weighting the models according to their ability to reproduce past observations may reduce the structural uncertainties Smith [2002a], IPCC [2013].

In addition, following the above comments and the suggestions made by some of the other reviewers, we tested our method by using one of the ensemble models as the “true” value and applied our method. This allowed us longer learning and validation periods, which include the different conditions predicted by the models. The results are presented in Figure 1 for one of the models in the CMIP5 serving as “observations.” The “observations” in this test were found to lay within the estimated uncertainty range. A similar performance was found when we used other models as the “observations.” In all cases, we found that our method reduced the uncertainties; still, in most cases, our forecast was under-confident, implying that the uncertainties could be further reduced.

Changes made:

- Clarified the statement.
- Extended the introduction and provided a referenced discussion of the problems with weighting projections and the advantages of weighting despite these problems.
- Added information on the out-of-sample validation and the reversed validation in the Methods section.

Figure 1: The 20-year running average of Global Mean Surface Temperature (GMST) change relative to the 1986–2005 average for the RCP scenarios included in the CMIP5. The thick lines represent the weighted ensemble mean for the 20-year running average GMST projections, and the shadings represent different significance levels (in the range of 0.1–0.9 with 0.1 steps) of the associated uncertainty (based on the EGA weighted ensemble and the AR estimation of the uncertainty ranges). The red line represents the “out-of-sample-model,” CSIRO-Mk3-6-0, which served as “observations” for this test of the suggested methodology (EGA+AR).

Reviewer #2:

The paper presents a study of a methodology to analyse ensemble projections with global climate models. It includes a method to generate a weighted mean of the ensembles and a method to link the ensemble spread to the uncertainties of the model mean. The weights are determined by using a sequential learning algorithm which minimizes the difference between the observations and the model mean. The relation between ensemble spread and the uncertainties is determined by a set of correction factors that are multiplied on the ensemble spread.

The methodology has been used before by the same authors in studies of decadal predictions. Here it is shown to produce reduced uncertainty estimates for the future climate projections.

The topic is important and the paper includes some interesting results.

However, I have some questions about the methodology that the authors should consider before I can recommend that the paper is accepted.

Major comments:

1) Validation: The validation period consists of 15 years following the training period. In this validation period the models have trends not too different from their trends in the training period.

This choice might give too optimistic results in the validation period. The authors should try to train the model on the period 1983-2017 and validate it on 1967-1982. Trends are different in these two periods and the results of the validation might change. Actually I don't understand why only data from 1967 are used. Why not use data from 1948 which would allow for longer training a validation periods?

Authors' Response: As mentioned in our manuscript, we used the 20-year running mean as the target variable. The value of the 20-year running average, assigned to each year, indicates the average GMST of the 20 years ending at the marked year. The data is available starting from 1948, and therefore, the first year for which we have the 20-year running average is 1967.

Following the reviewer's comment, we performed the suggested test of learning during the period of 1983–2017 and validating during the period of 1967–1982. We found that in this case, our methodology reduced the uncertainties but was not reliable (two upper panels of Figure 2). Extending the learning period to 40 years, i.e., learning from 1978–2017 and validating during the period from 1967–1977, resulted in a reduction of the errors and the uncertainty ranges, and improved the reliability compared with the equally weighted ensemble and the Gaussian assumption (two lower panels of Figure 2). Due to the different trends between the training and validation periods, as mentioned by the reviewer, our method required a longer training period in order to converge. Our results for the long-term projections are based on 50 years of learning, and therefore, we expect an even better performance of our algorithm in this case. Details of the additional tests that were performed in response to the other comments are provided in this letter and in the revised MS.

Changes made:

- Provided a clearer explanation of the 20-yr running average and clarified that the data from 1948 was used to generate the running average.
- Added information on the test in which the learning and validation periods were switched in the Methods section.

Perhaps some kind of cross-validation – using one model as the truth – could be used for validation.

Authors' Response: Following the reviewer's comment, we performed the suggested test using different models as the "observations." In almost all the cases we tested, we found that our algorithm reduced the errors, reduced the uncertainties, and improved the reliability of the forecast (compared with the simple average and the Gaussian approximation of the uncertainties). The results for one model, CSIRO-Mk3-6-0, which served as the "observation," are presented in Figure 1. The results show that despite the reduction of the uncertainties using our method, the forecast is still under-confident, and the actual uncertainties are smaller than those estimated. This finding is in agreement with our conclusion in the MS that there is a potential to further reduce the uncertainties. The results of this test and of a similar test using the CCSM4 as "observations" were added to the extended data.

Changes made: We added information on the out-of-sample experiment in the Methods section. In addition, two figures with the results of two different models (CSIRO-Mk3-6-0 and CCSM4) were added to the main manuscript as extended data figures.

Figure 2: Global mean surface temperature change relative to the 1986–2005 average. The thick black lines represent the ensemble mean, and the shadings represent the uncertainties with different significance levels (0.1–0.9). The thin black line represents the observations (NCEP reanalysis data). The two upper panels correspond to the learning period of 1983–2017 (35 years) and validation during the period of 1967–1982 (15 years). The two lower panels correspond to the learning period of 1978–2017 (40 years) and validation during the period of 1967–1977 (10 years). The left part of each panel (to the left of the dashed vertical line) represents the validation period, and the right part of each panel represents the learning period.

The sequential learning algorithm seems quite complex and it is not obvious what the algorithm actually learned. For example, does the last part of the training period influence the weights more than the first?

Authors' Response: In order to keep the MS concise and readable, we did not provide the previously published full details of the EGA. In the revised MS, we further elaborated on the rationale of the EGA while attempting to avoid too many details that may distract the reader. Specifically to the question regarding larger weights to more recent

observations, the algorithm does not consider recent observations to be more important. However, the learning is done in a sequential manner throughout the learning period. The weights are updated at each time step during this period in order to maximize the mutual information between the forecast and the observations. In previous works, we compared the EGA with different learning algorithms, as well as with an equally weighted ensemble and linear regression. We found that the EGA outperformed the other methods. We also illustrated the performance of the EGA and other learning algorithms using simple scenarios. These details were added to the revised MS with the relevant citations.

Changes made: Provided more details regarding the EGA upon its first mention.

It would also be interesting to see if the weights reflect how well the individual models reproduce the observed trend in the training period. The authors should also compare the results to a simpler method where the weights are determined by simply minimizing the mean square error over the training period.

Authors' Response: As mentioned above, we performed the tests suggested by the reviewer in previous works in which the algorithms were first introduced in the context of climate predictions. The weights are assigned in a sequential manner and, for the EGA, are based on the mutual information between the forecast and the observations. The EGA does not track the best model (i.e., the model with the smallest RMSE) but rather the observations. This is a clear advantage over other weighting schemes. The EGA was also shown to outperform linear regression and other methods. We elaborated more on these issues in the revised MS, and we refer the reader again to the previous works where those tests were performed. We believe that providing all this information in the current MS, which focuses on the reliability of the projections, would distract the reader.

Changes made: Provided more details regarding the weights assigned by the EGA.

2) I am somewhat confused about the influence of the “bias correction”. It seems that observed and modelled temperatures are all relative to their 1986-2005 averages. This means that the ensemble spread is low in this period. How would the results change if this interval was chosen differently? The CMIP5 historical experiments include all of the 20th century so there are many choices available.

Authors' Response: The anomaly (change in GMST) calculation was done with respect to the period of 1986–2005. The anomaly relative to this period is shown in order to enable comparison of our results with those reported in the IPCC [IPCC, 2013]. The EGA included a bias correction of the models (relative to the NCEP reanalysis data) based on the average error during the learning period. The details of the bias correction are provided in the MS.

3) The error on the model mean consists of both a bias and a variance. It is not obvious to me if the Asymmetric Range method includes the bias.

More generally, the relation between ensemble spread and the ensemble mean error has been studied in details in weather forecasts (see, e.g., the discussion in Christiansen 2019, doi: 10.1175/MWR-D-18-0211.1). There it seems that the relation is in general weak. Why should we expect it to be different for climate projections?

Authors' Response: The EGA includes a bias correction relative to the learning period (see the response to the previous comment). In addition, the quantity of interest in our work is the change in the GMST relative to the IPCC reference period of 1986–2005. Therefore, the bias is accounted for in our work.

We would also like to mention that in our work, regarding the uncertainties in decadal climate predictions [Strobach and Bel, 2017b], we demonstrated that considering a bias-corrected ensemble resulted in a clear relation between the ensemble spread and the error. The tests that we performed as part of this work on climate projections also showed that there is a relation between the two when the ensemble is bias-corrected. The work of Christiansen [2019] considers high dimensionality predictions that are different from the GMST projections. Specifically, Christiansen [2019] states: “Note that distance measures based on simple spatial averages, such as the difference between the modeled and the observed average Northern Hemisphere temperature, are not of high dimensionality and the results of this paper do therefore not hold for such diagnostics.”

The ensemble spread of weather predictions is likely to be dominated by internal variability while climate predictions and projections (where annual or longer term averages are considered) are dominated by model variability [Hawkins

and Sutton, 2009, Meehl et al., 2009, Strobach and Bel, 2017a]. This could be another reason for the difference between our analysis and the one mentioned by the reviewer.

Changes made:

- We added a brief discussion of the issue and included a citation of the work by Christiansen [2019] in the concluding sentences of the MS. “The method suggested and applied here was shown to be suitable for ensembles whose spread is dominated by model variability (rather than internal variability), which was found to be the case for multi-model ensembles of climate predictions and projections[Hawkins and Sutton, 2009, Meehl et al., 2009, Yokohata et al., 2013, Strobach and Bel, 2017a]. For ensembles of weather predictions where the internal variability is dominant, it was shown that there is a weak relation between the ensemble spread and its error[Christiansen, 2019].”

Minor comments:

page 3: Reliability could be better described.

Authors’ Response: We provided a better explanation of the term reliability in the context in which it was used in the MS.

Changes made:

- We added the following text: “The reliability is the correct quantification of the probability of the occurrence of different ranges of conditions Murphy [1973], Palmer et al. [2006], Leutbecher and Palmer [2008]. Specifically, we refer to reliability as the accurate quantification of the fraction of observations within different confidence level intervals (which is equivalent to comparing the cumulative probability distribution).”
- Following the comments of reviewer # 4, we extended the discussion of probabilistic forecast and the different scores that were previously suggested to measure its performance.

page 4: “These relatively short .. ”. Why is it important here that the response times are short?

Authors’ Response: The importance of the short response times is that they are reflected in changes to the GMST within the learning period. Slow responses, which would only be reflected by significant changes to the GMST decades or centuries after the triggering events, would not allow the study of the models ability to capture these responses because the observation period is limited.

Changes made: We added the following text to the relevant sentence, “These relatively short response times, which are reflected by the changes in the GMST within the period used for the evaluation of the model performances,...”

It would be nice if observations were also shown in Fig. 1 and Extended Data Figs. 1 and 2.

Authors’ Response: Following the reviewer’s suggestion, we added the observations to Figure 1. Extended Data Figure 1 showed the observations, but the line representing them was black, which may have made them hard to notice. We revised the figure, and the observations are now represented by a red line for better visibility. The caption was adjusted accordingly. A line representing the observations was also added to Extended Data Figure 2.

Changes made: Added the observations to all the relevant figures.

Page 14: p_t should be defined where it is first used.

Authors’ Response: We included the definition of p_t as the ensemble weighted mean.

Changes made: The sentence in the MS, “The PDs (probability distributions) of the GMST in different years differ due to the temporally varying STD, σ_t , and mean of the ensemble of GMST projections.” was replaced by “The PDs (probability distributions) of the GMST in different years differ due to the temporally varying STD, σ_t , and mean, p_t , of the ensemble of GMST projections. Note that the mean serves as the prediction of the ensemble, and both the mean

and the STD are based on the weights assigned to the ensemble members.”

Reviewer #3:

The submitted paper details a weighting methodology, presented here for time-series data, with the goal of producing calibrated forecasts from a multi-member ensemble of physical models. The method is applied to climate model data from the CMIP archive, producing a calibrated forecast for future scenarios with a significantly smaller uncertainty range than would be obtained by considering ensemble mean and standard deviation.

Unfortunately, I am not confident in the calibrated projections - for a number of reasons I detail below. The approach - which may have value in short-timescale initialized forecast problems does not address the key uncertainties present in longer timescale projections from a structurally diverse ensemble of models. The literature review also doesn't reflect the large amount of literature discussion on the topic of CMIP model weighting (and its limitations) over recent years. As a result, I cannot recommend this paper for publication in Nature Communications.

Authors' Response: Following the reviewer's comment, we added to the introduction a paragraph discussing the weighting of climate projections. We also provide citations to the major works demonstrating the advantage of weighting climate projections. Many works questioned the legitimacy of weighting projections [Kirtman et al., 2013, Weigel et al., 2010, Knutti et al., 2010, Sansom et al., 2013]. However, it was found that weighting climate projections improves the quality of the ensemble projections [Snape and Forster, 2014, Haughton et al., 2015, Abramowitz and Bishop, 2015, Gillett, 2015, Knutti et al., 2017, Sanderson et al., 2017, Borodina et al., 2017].

Changes made: We added citations and a discussion of the legitimacy of weighting climate projections. Specifically, we added the following text: “Evaluating climate projections by their past performances has been questioned, because the projections simulate future conditions that may be significantly different from past and current conditions, and certain limitations have been pointed out [Smith, 2002a, Kiehl, 2007, Weigel et al., 2010, Knutti et al., 2010, IPCC, 2013, Sansom et al., 2013]. However, numerous studies have shown that this approach is legitimate and indeed improves the quality of climate predictions and projections [IPCC, 2013, Snape and Forster, 2014, Gillett, 2015, Haughton et al., 2015, Abramowitz and Bishop, 2015, Strobach and Bel, 2015, 2016, Knutti et al., 2017, Sanderson et al., 2017, Borodina et al., 2017].”

The primary issue is related to the key assumptions - that there exists a time-independent multiplicative correction factor for the ensemble mean which applies in long term projections and that a model with smaller errors in global mean forced temperature response during the late 20th century is more reliable for century-scale projections than another model with larger errors. These statements are not self-evident, and have not been sufficiently demonstrated in an out-of-sample sense in this paper.

Authors' Response: We performed an out-of-sample test using several models as “observations” (see example in Figure 1). We also addressed the similarity between the models issue (mentioned below) by using as “out-of-sample” (“observations”) only models from modeling groups that have only one representative in our CMIP5 ensemble (i.e., CCSM4, CNRM-CM5, CSIRO-Mk3-6-0, CanESM2, EC-EARTH, and FGOALS-g2). The results demonstrate the validity and adequacy of our algorithm for the time scale spanned by the simulated projections. In all the tests, our method was shown to outperform the simple average with the Gaussian assumption method.

Changes made: We added a description of the out-of-sample experiment in the methods section and two additional figures showing the results of the test with two different models serving as “observations”.

Firstly, the real world is a single realization of internal variability (as the authors acknowledge), but the mitigating approach used (moving average 20 year means) is not sufficient to eliminate this problem - some modes of variability will project onto the 20 year moving average and cannot be cleanly separated from the forced response. Thus, an assumption that a single realization is the forced response will lead to bias in the weighted mean projection.

Authors' Response: The reviewer is correct to state that the natural variability of the climate system may be part of the change observed in the 20-year running average. However, this claim is true in general, and the uncertainties reported in the IPCC also include this component. The choice of a 20-year running average here was partially in order

to allow for an easy comparison with the last IPCC report IPCC [2013]. 20-year running averages are expected to be long enough to provide values dominated by the forced response. The natural variability may be large locally, but it is not expected to be dominant for large spatial means. This issue was already discussed in several papers (e.g., [Hawkins and Sutton, 2009]) and in the last IPCC report. In order to illustrate the ratio between the forced response and natural variability in climate projections, we show in Fig. 3 the projections of the ensemble models for the four RCPs considered. For RCP2.6 and RCP4.5, one can see that for each of the models, the projected natural variability is much smaller than the response to the projected atmospheric composition change (the fluctuations of the 20-yr GMST averages during the last projection period are much smaller than the change relative to the reference period). Even for RCP6.0 and RCP8.5, where the 20-yr average GMST still increases during the last period of the projections, the total change during the last 20 years is much smaller than the change relative to the reference period.

Figure 3: The 20-year running average of the mean surface temperature change relative to the 1986–2005 average for RCP2.6 (upper left), RCP4.5 (upper right), RCP6.0 (lower left), and RCP 8.5 (lower right). Colored lines represent different CMIP5 model projections, and the thick red line represents the NCEP reanalysis data. Three black lines represent the ensemble simple mean plus minus 1.96 STD.

Changes made: The revised MS includes the following statement: “The method suggested and applied here was shown to be suitable for ensembles whose spread is dominated by model variability (rather than internal variability), which was found to be the case for multi-model ensembles of climate predictions and projections [Hawkins and Sutton, 2009, Meehl et al., 2009, Yokohata et al., 2013, Strobach and Bel, 2017a].”

Secondly, even if the forced response was known - low errors in the historical record do not imply low errors in the future. The historical response is a combination of responses to changing greenhouse gas and other anthropogenic and natural forcings. A model might get the right answer for the wrong reasons if its greenhouse gas and aerosol sensitivities are both too large - such a model would have a low historical error but would be unreliable in the future (e.g. Kiehl 2007). The ratio of greenhouse gas and aerosol forcing in 2100 under RCP8.5, for example is vastly different to the present day.

Similarly, recent warming is a function of both radiative feedback response to forcing and ocean heat uptake - so

errors in these terms can also compensate and lead to the observed historical response in a model with compensating biases in climate sensitivity and ocean heat uptake (e.g. mauritzen 2017). These types of uncertainties (unlike for decadal predictions) are the first order uncertainty in future projections.

The legitimacy of weighting climate projections has indeed been debated by several authors for reasons including those mentioned by the reviewer. However, it is widely accepted that despite the caveats, it is legitimate and useful to weight the climate projections in order to reduce structural uncertainties [Knutti et al., 2013, IPCC, 2013, Snape and Forster, 2014, Gillett, 2015, Haughton et al., 2015, Abramowitz and Bishop, 2015, Strobach and Bel, 2015, 2016, Knutti et al., 2017, Sanderson et al., 2017, Borodina et al., 2017]. The paper by Kiehl [2007], which was mentioned by the reviewer, claims that there is a large uncertainty in climate projections regarding the anthropogenic forcing versus other factors. However, it concludes the following:

“It could also be argued that these results do not invalidate the application of climate models to projecting future climate for, at least, two reasons. First, within the range of uncertainty in aerosol forcing models have been benchmarked against the 20th century as a way of establishing a reasonable initial state for future predictions. The analogy would be to weather forecasting where models assimilate information to constrain the present state for improved prediction purposes. Climate models are forced within a range of uncertainty and yield a reasonable present state, which improves the models predictive capabilities. Second, many of the emission scenarios for the next 50 to 100 years indicate a substantial increase in greenhouse gases with associated large increase in greenhouse forcing. Given that the lifetime of these gases is orders of magnitude larger than that of aerosols, future anthropogenic forcing is dominated by greenhouse gases. Thus, the relative uncertainty in aerosol forcing may be less important for projecting future climate change.”

The ability of a model to reproduce past conditions does not guarantee its ability to correctly simulate future conditions; however, it is a necessary condition for models to be able to correctly reproduce past conditions [Smith, 2002b].

Changes made: In the revised MS, we included a referenced discussion of the legitimacy of weighting climate projections and the various approaches to this problem provided by previous works. We also elaborated on the structural uncertainties.

As such, representing the error around the mean as a time-invariant term is a strong assumption which has not been validated in the paper - and leads to significant overconfidence in the projected distributions. Rather, we would definitely expect the error term conditioned on global mean temperature alone to be time invariant, and to grow rapidly in the future - if we have conditioned the models based on recent climate, we have less reasons to trust our conditioning as the mix of forcing and rate of forcing change diverge from current values.

Authors' Response: Our methodology does not provide a time-invariant uncertainty range. The uncertainty range provided by the AR algorithm varies with time just like the ensemble spread (and mean) varies with time (in fact, it grows with time). However, we did find here, as found in previous studies, that a time-invariant factor multiplied by the ensemble spread (quantified by its STD) can reliably reproduce the uncertainty range. It was shown to outperform the equally weighted ensemble and the Gaussian assumption estimate of the uncertainties. Following the comments of the reviewer, and those of other reviewers, we also performed additional testing for validating the performance of our methodology for climate projections (as detailed throughout this letter).

Splitting the time-series into recent history, and pre-1995 periods is insufficient. The regime in terms of forcing mix, and rate of change of forcing is too similar during the training and validation periods and so the test is not out of sample.

The key tests which the authors would need to perform for an approach such as this (or any weighting strategy) would be an out of sample validation using 2100 temperatures from CMIP models as truth and predicting their future evolution using the remaining ensemble. This would allow an semi-objective test of confidence in an out of sample test (with the caveat that models with large similarities need to be removed from the comparison to make the test fair e.g. Sanderson 2017).

Authors' Response: We performed the out-of-sample test suggested by the reviewers. We considered several models as “observations” and validated the EGA+AR forecast for the entire simulated future period (the historical period was used for learning, just as was done when the NCEP reanalysis data was used as observations). We found that for

all the models that we used as “observations,” the EGA+AR outperformed the equally weighted ensemble combined with the Gaussian assumption. The EGA+AR forecast was more accurate (smaller RMSE) and more reliable (the reliability curve was closer to that of a perfectly reliable forecast).

Changes made: We included in the revised MS the information regarding the out-of-sample tests that further validate the adequacy of our method.

Kiehl, Jeffrey T. “Twentieth century climate model response and climate sensitivity.” *Geophysical Research Letters* 34.22 (2007).

Mauritsen, Thorsten, and Robert Pincus. “Committed warming inferred from observations.” *Nature Climate Change* 7.9 (2017): 652.

Sanderson, Benjamin M., Michael Wehner, and Reto Knutti. “Skill and independence weighting for multi-model assessments.” (2017): 2379-2395. APA

Reviewer #4:

I strongly support the eventual publication of this paper; it makes several points that are important and underappreciated both in the climate community and, just as importantly, in the community of numerate practitioners (say, economists) who wrongly assume just how much the properties of the model ensemble mean and ensemble standard deviation tell them. The paper also has some significant ambiguities or errors, I believe the paper should be appear after this a mended. I would strongly advise the editor NOT to reject this paper even give a highly negative review by another reviewer, please allow at least one revision before making a decision.

Wider Comments

The paper focuses on RMS and SD of GMT; it is now common to attempt fully probabilistic forecasts of annual GMT (Smith, L., Suckling, E., Thompson, E., Maynard, T. and Du, H. (2015) 'Towards improving the framework for probabilistic forecast evaluation', Climatic Change. I suggest that the authors note the probabilistic evaluation literature, and say (a few times) that exactly what they are doing (in scope and in evaluation).

Authors' Response: We thank the reviewer for the important comment. In the revised MS, we included a paragraph with a brief overview of previous works related to probabilistic forecast, and we also introduced our work in this context.

Changes made: The first part of the MS was revised to include a broader review of previous works and to place our work in the relevant context. Specifically, we added a referenced discussion of probabilistic forecast and the various scores that were suggested to measure the quality of such forecasts. We also explain that our work is situated within this category.

Perhaps the major items missing in the paper is the role of structure model error (see Smith, L.A. (2002) 'What might we learn from climate forecasts?', Proc. National Acad. Sci. USA, 4 (99): 2487-2492. and recent citations thereof) This paper is cited by the AR4 (pg 797) which suggests the climate distributions from ensembles are too narrow, as structural model error has not been properly taken into account. What assumptions to the authors make to bypass Smith (2002)'s claim?

Authors' Response: We accept the reviewer's comment. In the literature, the term “structural uncertainties” sometimes refers to what we described as the “model variability,” and in other works, they refer to the uncertainties due to un-modeled or inadequately modeled processes across models. We elaborated more on these uncertainties and the methods that were previously suggested in order to reduce these uncertainties, including weighting of the ensemble models.

Changes made: We added a paragraph discussing the structural uncertainties and the methods that were previously suggested in order to reduce them.

The two paragraphs above do not request more calculations, rather they require clarifying the assumptions required to apply the author's methods. This clarification is one the author strive for in their introduction and conclusion.

Defining the target as the 20 year average restricts out of sample forecasting in this century to one point in the near term (2020) with the next independent point at 2040.

Authors' Response: We clarified this point in the revised MS.

I would accept the paper after these minor corrections. I would ask to see and harsh negative reviews of this paper in any event.

Small detail:

pg 2 projection and prediction are NOT defined by their lead time but by what they are conditioned on, they are both Predictions (see IPCC glossary)

Authors' Response: We accept the comment, and in order to avoid distraction, we removed the details of the distinction between predictions and projections (the natural dynamics of the system vs. the response of the system to changes in the forcing).

Changes made: The first sentence of the MS was revised, and it now reads: "Climate predictions and projections are typically based on global circulation models, which simulate the multi-scale processes that form the climate system [IPCC, 2013]."

"The prediction of an ensemble is its weighted average ..." pg 3. in general this is simply false. Ensembles from their beginning (Lorenz and Tennekes) aimed to quantify variability as distributions, ideally probability forecasts; it is known that ensemble spread is a function of the initial model state. In climate models the mean state itself varies hugely within each CMIP (hence the taking of anomalies).

Authors' Response: We accept this comment and changed the sentence accordingly.

Changes made: The sentence was revised, and it now reads: "The prediction of an ensemble is the distribution of the climate variables or their anomalies. In particular, the expectation value of a variable anomaly is its weighted average (or the simple mean of the ensemble if the ensemble is equally weighted)."

pg 3 "much more reliable" please specify exactly what "more reliable" means (there are technical definitions for probability forecasts which are NOT what I think the authors mean) and which variables you claim it for.

Authors' Response: The term "reliability" was better defined in order to ensure that our meaning is clear.

Changes made: The sentence mentioned in the comment was modified, and it now reads: "The AR method was shown to provide much more reliable predictions (i.e., the fraction of measurements within a range estimated using the AR method was much closer to the expected fraction (for the entire range of expected fractions) than the fraction of measurements within a range estimated using the Gaussian estimation of the range) for the ensemble of CMIP5 decadal predictions of surface temperature and surface zonal wind [Strobach and Bel, 2017b]"

pg 4 Please clarify what is meant by "synchronized".

Authors' Response: We added a clarification of the term "synchronized" to the MS.

Changes made: The sentence in which we first used the term “synchronized” was revised, and it now reads: “Climate projections are not expected to be synchronized (in the sense that the timing of simulated fluctuations is the same as the timing in which they actually occur in the real climate systems) with the natural variability of the climate system (e.g., we do not expect the correct timing of future El Niño events).”

pg 4 Please clarify that by running averages the average is defined purely into the past.

Authors’ Response: We added a clarification in which we defined the 20-year running average.

Changes made: The sentence describing the 20-year running average was modified, and it now reads: “The simulated GMST 20-year running averages for the period of 1967–2100 were considered (the annual average GMST for the period 1948–2100 was used to construct the 20-year running average time series). The value assigned for each year represents the average GMST of the 20-year period ending on that year.”

pg 10 first full paragraph starting “Uncertainties in ... ” this is an important paragraph.

Authors’ Response: We thank the reviewer for the supportive comment.

References

- G. Abramowitz and C. H. Bishop. Climate model dependence and the ensemble dependence transformation of cmip projections. *Journal of Climate*, 28(6):2332–2348, 2015. doi: 10.1175/JCLI-D-14-00364.1.
- A. Borodina, E. M. Fischer, and R. Knutti. Emergent constraints in climate projections: A case study of changes in high-latitude temperature variability. *Journal of Climate*, 30(10):3655–3670, 2017. doi: 10.1175/JCLI-D-16-0662.1.
- B. Christiansen. Analysis of ensemble mean forecasts: The blessings of high dimensionality. *Monthly Weather Review*, 147(5):1699–1712, 2019. doi: 10.1175/MWR-D-18-0211.1.
- N. P. Gillett. Weighting climate model projections using observational constraints. *Philosophical Transactions of the Royal Society A: Mathematical, Physical and Engineering Sciences*, 373(2054):20140425, 2015. doi: 10.1098/rsta.2014.0425.
- N. Haughton, G. Abramowitz, A. Pitman, and S. J. Phipps. Weighting climate model ensembles for mean and variance estimates. *Climate Dynamics*, 45(11):3169–3181, Dec 2015. ISSN 1432-0894. doi: 10.1007/s00382-015-2531-3.
- E. Hawkins and R. Sutton. The Potential to Narrow Uncertainty in Regional Climate Predictions. *Bulletin of the American Meteorological Society*, 90(8):1095–1107, Aug. 2009. ISSN 0003-0007. doi: 10.1175/2009BAMS2607.1.
- IPCC. *Climate Change 2013: The Physical Science Basis. Contribution of Working Group I to the Fifth Assessment Report of the Intergovernmental Panel on Climate Change*. Cambridge University Press, Cambridge, United Kingdom and New York, NY, USA, 2013. ISBN ISBN 978-1-107-66182-0.
- J. T. Kiehl. Twentieth century climate model response and climate sensitivity. *Geophysical Research Letters*, 34(22), 2007. doi: 10.1029/2007GL031383.
- B. Kirtman, S. B. Power, J. A. Adedoyin, G. Boer, R. Bojariu, I. Camilloni, F. J. Doblas-Reyes, A. M. Fiore, M. Kimoto, G. A. Meehl, M. Prather, A. Sarr, C. Schr, R. Sutton, G. J. van Oldenborgh, G. Vecchi, and H. J. Wang. Near-term climate change: Projections and predictability. In T. F. Stocker, D. Qin, G. K. Plattner, M. Tignor, S. K. Allen, J. Boschung, A. Nauels, Y. Xia, V. Bex, and P. Midgley, editors, *Climate Change 2013: The Physical Science Basis. Contribution of Working Group I to the Fifth Assessment Report of the Intergovernmental Panel on Climate Change*. Cambridge University Press, Cambridge, United Kingdom and New York, NY, USA, 2013.
- R. Knutti, R. Furrer, C. Tebaldi, J. Cermak, and G. A. Meehl. Challenges in combining projections from multiple climate models. *Journal of Climate*, 23(10):2739–2758, 2010.

- R. Knutti, D. Masson, and A. Gettelman. Climate model genealogy: Generation cmip5 and how we got there. *Geophysical Research Letters*, 40(6):1194–1199, 2013. doi: 10.1002/grl.50256.
- R. Knutti, J. Sedlek, B. M. Sanderson, R. Lorenz, E. M. Fischer, and V. Eyring. A climate model projection weighting scheme accounting for performance and interdependence. *Geophysical Research Letters*, 44(4):1909–1918, 2017. doi: 10.1002/2016GL072012.
- M. Leutbecher and T. Palmer. Ensemble forecasting. *Journal of Computational Physics*, 227(7):3515 – 3539, 2008. ISSN 0021-9991. doi: <https://doi.org/10.1016/j.jcp.2007.02.014>. Predicting weather, climate and extreme events.
- G. A. Meehl, L. Goddard, J. Murphy, R. J. Stouffer, G. Boer, G. Danabasoglu, K. Dixon, M. A. Giorgetta, A. M. Greene, E. Hawkins, G. Hegerl, D. Karoly, N. Keenlyside, M. Kimoto, B. Kirtman, A. Navarra, R. Pulwarty, D. Smith, D. Stammer, and T. Stockdale. Decadal prediction. *Bulletin of the American Meteorological Society*, 90(10):1467–1486, 2009. doi: 10.1175/2009BAMS2778.1.
- A. H. Murphy. A new vector partition of the probability score. *Journal of Applied Meteorology*, 12(4):595–600, 1973. doi: 10.1175/1520-0450(1973)012<0595:ANVPOT>2.0.CO;2.
- T. Palmer, R. Buizza, R. Hagedorn, A. Lawrence, M. Leutbecher, and L. Smith. Ensemble prediction: A pedagogical perspective. *ECMWF Newsletter*, 106:10–17, 2006.
- B. M. Sanderson, M. Wehner, and R. Knutti. Skill and independence weighting for multi-model assessments. *Geoscientific Model Development*, 10(6):2379–2395, 2017. doi: 10.5194/gmd-10-2379-2017.
- P. G. Sansom, D. B. Stephenson, C. A. T. Ferro, G. Zappa, and L. Shaffrey. Simple uncertainty frameworks for selecting weighting schemes and interpreting multimodel ensemble climate change experiments. *Journal of Climate*, 26(12):4017–4037, 2013. doi: 10.1175/JCLI-D-12-00462.1.
- L. A. Smith. What might we learn from climate forecasts? *Proceedings of the National Academy of Sciences*, 99(suppl 1):2487–2492, 2002a. ISSN 0027-8424. doi: 10.1073/pnas.012580599.
- L. A. Smith. What might we learn from climate forecasts? *Proceedings of the National Academy of Sciences*, 99(suppl 1):2487–2492, 2002b. ISSN 0027-8424. doi: 10.1073/pnas.012580599.
- T. J. Snape and P. M. Forster. Decline of arctic sea ice: Evaluation and weighting of cmip5 projections. *Journal of Geophysical Research: Atmospheres*, 119(2):546–554, 2014. doi: 10.1002/2013JD020593.
- E. Strobach and G. Bel. Improvement of climate predictions and reduction of their uncertainties using learning algorithms. *Atmospheric Chemistry and Physics*, 15:8631–8641, 2015. doi: 10.5194/acp-15-8631-2015.
- E. Strobach and G. Bel. Decadal climate predictions using sequential learning algorithms. *Journal of Climate*, 0(0):null, 2016. doi: 10.1175/JCLI-D-15-0648.1.
- E. Strobach and G. Bel. The contribution of internal and model variabilities to the uncertainty in cmip5 decadal climate predictions. *Climate Dynamics*, Mar 2017a. ISSN 1432-0894. doi: 10.1007/s00382-016-3507-7.
- E. Strobach and G. Bel. Quantifying the uncertainties in an ensemble of decadal climate predictions. *Journal of Geophysical Research: Atmospheres*, 122(24):13,191–13,200, 2017b. ISSN 2169-8996. doi: 10.1002/2017JD027249. 2017JD027249.
- A. P. Weigel, R. Knutti, M. A. Liniger, and C. Appenzeller. Risks of model weighting in multimodel climate projections. *Journal of Climate*, 23(15):4175–4191, 2010.
- T. Yokohata, J. D. Annan, M. Collins, C. S. Jackson, H. Shiogama, M. Watanabe, S. Emori, M. Yoshimori, M. Abe, M. J. Webb, and J. C. Hargreaves. Reliability and importance of structural diversity of climate model ensembles. *Climate Dynamics*, 41(9):2745–2763, Nov 2013. ISSN 1432-0894. doi: 10.1007/s00382-013-1733-9.

Reviewers' comments:

Reviewer #2 (Remarks to the Author):

I think that the authors have addressed my comments in a satisfactory way and I will now suggest that the paper is accepted for publication.

Minor comments:

The results of the cross-validation test should be presented more systematic and not just by showing results from a few tests. What is the average result when using all models as 'observations' and what is the spread?

I also think the results should be mentioned in the main part of the paper and not only in the methods section. This also goes for the 'reversed' experiment. It seems to me that the 'reversed' experiment might be more relevant -- because it includes a different trend than in the calibration period -- than when the first period is used for training

You will probably need to include a formal acknowledgement of CMIP5. See <https://pcmdi.llnl.gov/mips/cmip5/citation.html>

In the list of references the abbreviations CMIP and NCEP are sometimes not capitalized.

Reviewer #3 (Remarks to the Author):

Thanks to the authors for the revised manuscript and the response to the original reviews. However, I still find this paper to be a highly overconfident representation of projection uncertainties - and unsuitable for publication in its current form.

The authors' headline result suggests that weighting projections by recent temperature trends can reduce end of century global mean temperature uncertainty by a factor of 3 to 5 (Figure 3, bottom panels). If true, this would be revolutionary, a near order magnitude difference in confidence in projections compared with IPCC AR5 estimates.

Skill weighting remains highly contentious - notably the IPCC has explicitly rejected implementing any form of weighting in its assessments, and the publications listed in this submission do not defend the idea that weighting in a general sense is "legitimate and indeed improves the quality of climate predictions and projections", rather in some cases they outline specific cases where it might be appropriate. The sea-ice projection in Knutti 2017, for example - is a specific case where weighting is appropriate because many models over- or under- estimate the historical sea-ice extent. In other cases where models are weighted in a more general sense (Sanderson 2017, for example) - the benefits of skill weighting are marginal and debatable. In cases where global mean temperature timeseries were used to constrain future behavior (Gillett 2015, Abramowitz and Bishop 2015) - the published uncertainties in future projections were much greater than in the present study.

There is also a philosophical problem with weighting by the transient evolution of global mean temperature - a value which we know is highly tuned in the climate model development process as groups endeavor to re-produce 20th century temperature evolution. As such - this is not independent information, it is information which has already been used in development as a tuning target.

We also know (Rose 2014, Rungenstein 2016, Andrews 2018) that the transient evolution of temperatures is a poor predictor of equilibrium response under concentration stabilization - yet the

results here suggest that RCP2.6 stabilization temperatures can be highly constrained by 20th century temperature trends (Figure 3, lower panel). This result is at odds with present understanding of the long term dynamics of the climate system.

All of this implies that robust validation of the long term skill of the technique is imperative for this study. As I noted in the first review - a test of weighting skill using adjacent time periods (1967-2005 for training and 2006-2017 for validation) is not a meaningful test of long term projections, and the currently implemented perfect-model long timescale cross-validation consists of only two models - one of which (CCSM4) has a number of near replicates in the ensemble (NorESM1-M, NorESM1-ME, CESM1-BGC) - thus rendering the test unfair. An out of sample validation needs to be a comprehensive assessment of the ability of the method to constrain long term projections, not simply a couple of examples. Notably, even in the authors' examples in the case of the CSIRO model (Extended Figure 5) show that the weighting scheme over-confidently suggests temperatures will stabilize in RCP2.6 where the model does not.

A meaningful cross-validation would be a plot of projected end of century probability ranges for each model in the ensemble treated as truth as a function of its actual warming for each RCP - performed once for all models in the ensemble, with all highly related models removed from the predictive ensemble.

Rose, Brian EJ, et al. "The dependence of transient climate sensitivity and radiative feedbacks on the spatial pattern of ocean heat uptake." *Geophysical Research Letters* 41.3 (2014): 1071-1078.

Rugenstein, Maria AA, Ken Caldeira, and Reto Knutti. "Dependence of global radiative feedbacks on evolving patterns of surface heat fluxes." *Geophysical Research Letters* 43.18 (2016): 9877-9885.

Andrews, Timothy, et al. "Accounting for changing temperature patterns increases historical estimates of climate sensitivity." *Geophysical Research Letters* 45.16 (2018): 8490-8499.

We thank both reviewers for the careful assessment of our manuscript and for the constructive comments. Our point-by-point responses are provided below. For clarity, we used italicized font for the reviewers' comments, normal blue font for our responses, and normal green font for the description of the changes to the MS.

Reviewer #2:

I think that the authors have addressed my comments in a satisfactory way and I will now suggest that the paper is accepted for publication.

Minor comments:

The results of the cross-validation test should be presented more systematic and not just by showing results from a few tests. What is the average result when using all models as 'observations' and what is the spread?

Authors' Response: As requested by the reviewer, we performed the cross-validation test for each of the ensemble models. The results of the analysis support the results presented in the previous version of the MS. We found that the error and the spread are significantly reduced by our method. The reliability is significantly reduced for RCP2.6 and RCP6.0 and non-significantly for RCP4.5 and RCP8.5. However, even if the reliability were not significantly improved, the spread is smaller without affecting the reliability, thereby supporting the suggested methodology.

Changes made: Two more figures were added to the extended data. One figure is a graphical representation of the closely related models, which in turn were removed from the ensemble for the cross-validation test in which the model was used as the "true value." In the second figure, the scores for each model and the RCP, together with the mean and the spread, were included. Additional information describing the analysis is included in the Methods section.

I also think the results should be mentioned in the main part of the paper and not only in the methods section. This also goes for the 'reversed' experiment. It seems to me that the 'reversed' experiment might be more relevant – because it includes a different trend than in the calibration period – than when the first period is used for training.

Authors' Response: We agree with the reviewer that the 'reversed' experiment is more relevant. Although it is shorter, it has a different trend. Also, in the 'reversed' experiment, we try to learn the true time series of the variable that the models are trying to project. In the out-of-sample experiment, we try to optimize the ensemble projections to another "verification model." Some of the models do not represent the actual climate dynamics, and therefore, their use for verification purposes is limited in scope.

Changes made: We added information on the 'reversed' experiment and the out-of-sample test in the main manuscript with references to the Methods section for detailed descriptions of these tests.

You will probably need to include a formal acknowledgement of CMIP5. See <https://pcmdi.llnl.gov/mips/cmip5/citation.html>

Authors' Response: A formal acknowledgement of CMIP5 is already included in the original text in the Acknowledgements section.

In the list of references the abbreviations CMIP and NCEP are sometimes not capitalized.

Changes made: We corrected these typos.

Reviewer #3:

Thanks to the authors for the revised manuscript and the response to the original reviews. However, I still find this paper to be a highly overconfident representation of projection uncertainties - and unsuitable for publication in its current form.

The authors' headline result suggests that weighting projections by recent temperature trends can reduce end of century global mean temperature uncertainty by a factor of 3 to 5 (Figure 3, bottom panels). If true, this would be

revolutionary, a near order magnitude difference in confidence in projections compared with IPCC AR5 estimates.

Skill weighting remains highly contentious - notably the IPCC has explicitly rejected implementing any form of weighting in its assessments, and the publications listed in this submission do not defend the idea that weighting in a general sense is "legitimate and indeed improves the quality of climate predictions and projections", rather in some cases they outline specific cases where it might be appropriate. The sea-ice projection in Knutti 2017, for example - is a specific case where weighting is appropriate because many models over- or under- estimate the historical sea-ice extent. In other cases where models are weighted in a more general sense (Sanderson 2017, for example) - the benefits of skill weighting are marginal and debatable. In cases where global mean temperature timeseries were used to constrain future behavior (Gillett 2015, Abramowitz and Bishop 2015) - the published uncertainties in future projections were much greater than in the present study.

There is also a philosophical problem with weighting by the transient evolution of global mean temperature - a value which we know is highly tuned in the climate model development process as groups endeavor to re-produce 20th century temperature evolution. As such - this is not independent information, it is information which has already been used in development as a tuning target.

We also know (Rose 2014, Rungenstein 2016, Andrews 2018) that the transient evolution of temperatures is a poor predictor of equilibrium response under concentration stabilization - yet the results here suggest that RCP2.6 stabilization temperatures can be highly constrained by 20th century temperature trends (Figure 3, lower panel). This result is at odds with present understanding of the long term dynamics of the climate system.

All of this implies that robust validation of the long term skill of the technique is imperative for this study. As I noted in the first review - a test of weighting skill using adjacent time periods (1967-2005 for training and 2006-2017 for validation) is not a meaningful test of long term projections, and the currently implemented perfect-model long timescale cross-validation consists of only two models - one of which (CCSM4) has a number of near replicates in the ensemble (NorESM1-M, NorESM1-ME, CESM1-BGC) - thus rendering the test unfair. An out of sample validation needs to be a comprehensive assessment of the ability of the method to constrain long term projections, not simply a couple of examples. Notably, even in the authors' examples in the case of the CSIRO model (Extended Figure 5) show that the weighting scheme over-confidently suggests temperatures will stabilize in RCP2.6 where the model does not.

A meaningful cross-validation would be a plot of projected end of century probability ranges for each model in the ensemble treated as truth as a function of its actual warming for each RCP - performed once for all models in the ensemble, with all highly related models removed from the predictive ensemble.

Rose, Brian EJ, et al. "The dependence of transient climate sensitivity and radiative feedbacks on the spatial pattern of ocean heat uptake." *Geophysical Research Letters* 41.3 (2014): 1071-1078.

Rugenstein, Maria AA, Ken Caldeira, and Reto Knutti. "Dependence of global radiative feedbacks on evolving patterns of surface heat fluxes." *Geophysical Research Letters* 43.18 (2016): 9877-9885.

Andrews, Timothy, et al. "Accounting for changing temperature patterns increases historical estimates of climate sensitivity." *Geophysical Research Letters* 45.16 (2018): 8490-8499.

Authors' Response: As the reviewer summarized, the main request is for a full and detailed out-of-sample test in which related models are removed from the projecting ensemble. We performed the cross-validation test for each of the ensemble models. The results of the analysis support the results presented in the previous version of the MS. We found that the error and the spread are significantly reduced. The reliability is significantly reduced for RCP2.6 and RCP6.0 and non-significantly for RCP4.5 and RCP8.5. More details are included in the Methods section. It is important to note that despite the non-significant improvement of the reliability for RCP4.5 and RCP8.5, the reduced uncertainties with the similar reliability are an improvement to the state-of-the-art.

Changes made: Two more figures were added to the extended data. One figure is a graphical representation of the closely related models, which in turn were removed from the ensemble for the cross-validation test in which the model was used as the "true value." In the second figure, the scores for each model and the RCP, together with the mean and the spread, were included. Additional information describing the analysis is included in the Methods section.

Reviewers' comments:

Reviewer #3 (Remarks to the Author):

Thanks to the authors for the addition of the out-of-sample test.

However - various minor aspects remain unclear in the revised manuscript, which make it difficult to assess the results.

1 - Please make clear what time periods are being used for extended figure 5. I remain concerned that the years directly following the training period are not independent of the training data - and a more robust assessment of the technique would be to compute skill scores on late 21st century warming.

2 - Ext Figure 5 is not a very clear assessment of out of sample skill. It is not apparent to what degree the prediction is degraded by the exclusion of related models. The integrated Reliability score also makes it impossible to assess whether the tails of the predicted distribution are well represented.

A clear, simple, demonstration of the skill of the technique in the out of sample test would be, for example, to show an errorplot with projected weighted temperature distributions in 2070-2100 as a function of actual model 2070-2100 warming in each CMIP5 model, for each scenario.

2 - Detail exactly how the reliability score was computed in the out of sample test.

3 - The result that the RCP2.6 is more predictable than RCP8.5 is counterintuitive, given that both RCP8.5 21st century warming and recent warming trends are both to some degree correlated with metrics of climate sensitivity, while RCP2.6 is not (see Grose et al, 2017). It warrants discussion why the authors find a different result.

Grose, Michael R., et al. "What Climate Sensitivity Index Is Most Useful for Projections?." *Geophysical Research Letters* 45.3 (2018): 1559-1566.
<https://agupubs.onlinelibrary.wiley.com/doi/abs/10.1002/2017GL075742>

Reviewer #3:

We appreciate the reviewer's effort to assess the reliability of our manuscript. Our point-by-point responses are provided below. For clarity, we used italicized font for the reviewer's comments, normal blue font for our responses, and normal green font for the description of the changes to the MS.

Thanks to the authors for the addition of the out-of-sample test.

However - various minor aspects remain unclear in the revised manuscript, which make it difficult to assess the results.

1 - Please make clear what time periods are being used for extended figure 5. I remain concerned that the years directly following the training period are not independent of the training data - and a more robust assessment of the technique would be to compute skill scores on late 21st century warming.

Authors' Response: *Following the reviewer's comment, we revised the assessment of the skill to the years 2040–2099; all the time points in the validation time series are independent of the data used for training. In the previous version, we considered the period of 2020–2099. We provide here separate figures for the periods of 2040–2069 (Fig. 1) and 2070–2099 (Fig. 2). The results show that the differences between the two periods are not considerable. The most apparent difference is the reduced reliability of our method for RCP2.6 in the latest period. We find it more informative to include in the MS the results for the entire validation period.*

2 - Ext Figure 5 is not a very clear assessment of out of sample skill. It is not apparent to what degree the prediction is degraded by the exclusion of related models. The integrated Reliability score also makes it impossible to assess whether the tails of the predicted distribution are well represented.

A clear, simple, demonstration of the skill of the technique in the out of sample test would be, for example, to show an error plot with projected weighted temperature distributions in 2070-2100 as a function of actual model 2070-2100 warming in each CMIP5 model, for each scenario.

Authors' Response: *We believe that the scores provided in the main manuscript contain the important information needed to assess the skill of our methodology. It is important to note that the main point of the MS is not the validation of the method, but rather the implications of its validity (the results derived using the method). Therefore, distractions from the main theme should be kept to the required minimum. However, as requested by the reviewer, we provide here the error plots for the beginning and the end of the independent projections period: 2040–2069 and 2070–2099 (Figures 3 and 4). In the two figures, most of the observations fall inside our range of uncertainty, which is also smaller than the range derived using the equally weighted ensemble and the Gaussian assumption (in most cases). In addition, the error of the weighted ensemble is smaller than the error of the equally weighted ensemble. From these two figures, it is difficult to infer the qualitative difference between the performances of our method in the two periods. The results are provided for 22 models (excluding the groups of models that consistently predict the highest and lowest GMST). The reason for the exclusion of the extreme models is the fact that methods based on the weighted (including the simple average) ensemble are not capable of predicting values that are outside the range spanned by the ensemble members.*

3 - Detail exactly how the reliability score was computed in the out of sample test.

Authors' Response: *As explained in the main manuscript, "we used the mean squared deviation of the reliability curve from the identity line (representing a perfectly reliable forecast) as a reliability score." This is basically the Ranked Probability Score (RPS).*

4 - The result that the RCP2.6 is more predictable than RCP8.5 is counterintuitive, given that both RCP8.5 21st century warming and recent warming trends are both to some degree correlated with metrics of climate sensitivity, while RCP2.6 is not (see Grose et al, 2017). It warrants discussion why the authors find a different result.

Grose, Michael R., et al. "What Climate Sensitivity Index Is Most Useful for Projections?." Geophysical Research Letters 45.3 (2018): 1559-1566. <https://agupubs.onlinelibrary.wiley.com/doi/abs/10.1002/2017GL075742>

Authors' Response: *We appreciate the reviewer's comment, but we believe that this kind of discussion would re-*

Figure 1: **Left panels:** Relative RMSE ($\frac{RMSE_{EGA+AR} - RMSE_{AVG}}{(RMSE_{EGA+AR} + RMSE_{AVG})/2}$), relative 90 percentile range ($\frac{RAN_{EGA+AR} - RAN_{AVG}}{(RAN_{EGA+AR} + RAN_{AVG})/2}$), and relative reliability ($\frac{REL_{AVG} - REL_{EGA+AR}}{(REL_{EGA+AR} + REL_{AVG})/2}$) for the 24 models (the results for each model correspond to the out-of-sample test in which this model was used as “observation”) and four RCPs. **Right panels:** A summary of the results presenting the mean and the 90 percentile confidence interval (error bars).. The data correspond to the period of 2040–2069.

quire elaborating on the climate sensitivity index and its relation to the quality of projections. This type of assessment and discussion are beyond the scope of the current manuscript.

Figure 2: **Left panels:** Relative RMSE ($\frac{RMSE_{EGA+AR} - RMSE_{AVG}}{(RMSE_{EGA+AR} + RMSE_{AVG})/2}$), relative 90 percentile range ($\frac{RAN_{EGA+AR} - RAN_{AVG}}{(RAN_{EGA+AR} + RAN_{AVG})/2}$), and relative reliability ($\frac{REL_{AVG} - REL_{EGA+AR}}{(REL_{EGA+AR} + REL_{AVG})/2}$) for the 24 models (the results for each model correspond to the out-of-sample test in which this model was used as “observation”) and four RCPs. **Right panels:** A summary of the results presenting the mean and the 90 percentile confidence interval (error bars). The data correspond to the period of 2070–2099.

2040-2069

Figure 3: Error plot of the GMST change for the specified 24 models and four RCPs (from top to bottom: RCP2.6, RCP4.5, RCP6.0, RCP8.5). The circles, boxes, and error bars represent the ensemble mean, the 25%–75% uncertainty range, and the 5%–95% uncertainty range, respectively. The blue, red, and green colors correspond to the verification model, the estimation based on the EGA and AR methods, and the estimation based on the equally weighted ensemble and the Gaussian assumption, respectively. The data correspond to the period of 2040–2069.

2070-2099

Figure 4: Error plot of the GMST change for the 24 models and four RCPs (from top to bottom: RCP2.6, RCP4.5, RCP6.0, RCP8.5). The circles, boxes, and error bars represent the ensemble mean, the 25%–75% uncertainty range, and the 5%–95% uncertainty range, respectively. The blue, red, and green colors correspond to the verification model, the estimation based on the EGA and AR methods, and the estimation based on the equally weighted ensemble and the Gaussian assumption, respectively. The data correspond to the period of 2070–2099.

REVIEWERS' COMMENTS:

Reviewer #3 (Remarks to the Author):

Thanks to the authors for their extensive further investigation of their results and accommodation of the review requests. The results are surprising, but the out-of-sample results seem robust - and given that, the paper is worthy of publication and discussion in the wider community.

There still remain questions as to why historical temperatures appear to be providing such a strong constraint, but I accept for the purposes of this publication it is sufficient to note that the constraint exists. I have no further issues to address for publication.